# Humanized Ovarian Cancer Patient-Derived Xenografts for Improved Preclinical Evaluation of Immunotherapies

**DOI:** 10.3390/cancers14133092

**Published:** 2022-06-23

**Authors:** Katrin Kleinmanns, Stein-Erik Gullaksen, Geir Bredholt, Ben Davidson, Cecilie Fredvik Torkildsen, Sindre Grindheim, Line Bjørge, Emmet McCormack

**Affiliations:** 1Department of Clinical Science, Centre for Cancer Biomarkers CCBIO, University of Bergen, 5020 Bergen, Norway; stein.gullaksen@uib.no (S.-E.G.); geir.bredholt@uib.no (G.B.); cecilie.torkildsen@uib.no (C.F.T.); sindre.grindheim@helse-bergen.no (S.G.); line.bjorge@uib.no (L.B.); 2Department of Pathology, Oslo University Hospital, Norwegian Radium Hospital, 0310 Oslo, Norway; ben.davidson@medisin.uio.no; 3Faculty of Medicine, Institute of Clinical Medicine, University of Oslo, 0316 Oslo, Norway; 4Department of Obstetrics and Gynecology, Stavanger University Hospital, 4011 Stavanger, Norway; 5Department of Obstetrics and Gynecology, Haukeland University Hospital, 5021 Bergen, Norway; 6Vivarium, Department of Clinical Science, University of Bergen, Jonas Lies vei 65, 5021 Bergen, Norway; 7Centre for Pharmacy, Department of Clinical Science, The University of Bergen, Jonas Lies vei 65, 5021 Bergen, Norway

**Keywords:** high-grade serous ovarian cancer, orthotopic mouse models, mass cytometry, NSG, NSGS, nivolumab, PD-1, hu PDX

## Abstract

**Simple Summary:**

Epithelial ovarian cancer (EOC) is a heterogenous disease and new combination therapies are employed to improve treatment, decrease disease recurrence, and avoid the development of treatment resistance. Immunotherapy has been suggested to boost the immune system and improve the prognosis of EOC patients. However, overall low response rates and missing reliable biomarkers to stratify patients to their best-suited personalized treatment regime hinder the successful implementation. Our aim is to advance the development and characterization of humanized patient-derived xenograft models aiding to unravel the function and interaction of the unique tumor microenvironment and the immune system in EOC. These developed and clinically relevant humanized models of EOC have the potential to test various immune cell-targeting combination therapies and identify mechanisms in heterogenous EOC cohorts to ultimately allow patient stratification.

**Abstract:**

High-grade serous ovarian cancer (HGSOC) has poor prognosis and new treatment modalities are needed. Immunotherapy, with checkpoint inhibitors, have demonstrated limited impact. To evaluate the suitability for immunotherapeutics, contextualized preclinical models are required to secure meaningful clinical translation. Therefore, we developed and characterized humanized patient-derived xenograft (hu PDX) murine models of HGSOC, which were established by orthotopic implantation of tumor cell suspensions and intravenous injection of CD34^+^ cells isolated from umbilical cord blood samples. The developing human immune system in NSG and NSGS mice was followed longitudinally by flow cytometry and characterized by mass cytometry with a panel of 34 surface markers. Molecular imaging of tumor burden, survival analysis, and characterization of tumor-infiltrating immune cells was performed to assess the treatment response to anti-PD-1 (nivolumab) monotherapy. Successful generation of hu PDX models was achieved. Mice treated with nivolumab showed a decrease in tumor burden, however no significant survival benefit was identified when compared to untreated controls. No correlation was seen between PD-L1 expression and CD8 T cell infiltration and response parameters. As the characterization showed an immune infiltration of predominantly myeloid cells, similar to what is observed in HGSOC patients, the models may have the potential to evaluate the importance of myeloid cell immunomodulation as well.

## 1. Introduction

Patients diagnosed with the most common and most aggressive subtype of epithelial ovarian cancer (EOC), high-grade serous ovarian carcinoma (HGSOC), exhibit an initial encouraging response to aggressive frontline treatment with debulking surgery and chemotherapy. However, most patients will experience recurrence, leading to a 5-year survival of less than 50% [1]. With each recurrence, the response to treatment decreases and treatment resistance becomes a common trademark [2]. Following the successful implementation of targeted therapies into the treatment regimen for HGSOC, immunotherapy has attracted significant interest, including immune checkpoint inhibitors (ICI) [3]. ICI can yield impressive clinical responses, earning FDA approvals for other solid tumors. HGSOC is classified as an immunogenic tumor with an intermediate mutational load [4]. It was the first tumor identified where an increased density of intraepithelial tumor-infiltrating lymphocytes (TIL) at diagnosis correlated with prolonged overall survival (OS) [5,6]. Exhausted programmed death 1 (PD-1^+^) TIL are believed to be reinvigorated by ICI, with a greater treatment response for exhausted tumor-specific TIL and dendritic cell positive tumors [7]. Therefore, immunotherapy has been suggested to have the potential to prevent or delay recurrence and increase the survival rate in patients. However, clinical trials investigating anti-PD-1 (nivolumab) monotherapies reported low overall response rates of around 15% [8], which are in agreement with other ICI tested in EOC (10% ORR; Javelin [9] Keynote-100 [10]). Even though the identified immune-reactive molecular subtype of HGSOC showed significantly longer OS, immunosuppressive networks in the ovarian tumor microenvironment (TME), such as regulatory T cells (Treg), indoleamine 2,3-dioxygenase (IDO), IL-10, programmed death-ligand (PD-L1), myeloid cells and vascular endothelial growth factor (VEGF), are associated with tumor progression and poor prognosis, thus impeding the efficacy of immunotherapy [11,12]. To overcome low response rates, the complexity of the immunological response must be understood to identify the biomarkers that predict treatment efficacy and to stratify patients to personalized combination therapies.

Despite advances towards personalized precision medicine, particularly in the immuno-oncology setting, there is a paucity of relevant animal models to truly interrogate the preclinical concepts prior to clinical translation [13]. Innovative mouse models are critical for deciphering the underlying immune mechanisms and immunosuppressive pathways. Such information will enable optimization of treatment regimens. Substantial efforts have been devoted to generating humanized mouse models, whereby immunodeficient mice are engrafted with hematopoietic CD34^+^ cells that differentiate into a functional human immune system. The immunodeficient NOD-*scid* IL2rγ^null^ (NSG) mouse strain allows multilineage differentiation and is the most frequently used strain for human immune reconstitution [14,15]. Bone marrow preconditioning before the injection of stem cells by whole-body irradiation or injection of the alkylating agent busulfan creates a niche for the differentiation of human hematopoietic CD34^+^ cells. However, several essential cytokines do not cross-react between species, obstructing differentiation and maturation of several immune cell types. Thus, NSG-SGM3 (NSGS) mice have been developed that express supraphysiological concentrations (>200 pg/mL) of the human hematopoietic cytokines SCF, GM-CSF and IL-3, which support the engraftment of leukemic cells and the generation of a competent human immune system [16,17]. This results in improved human myelopoiesis, Treg and natural killer (NK) cell development, albeit with decreased erythropoiesis compared to NSG mice [18,19]. Since NSGS mice develop severe anemia after humanization, the use of the model for experimental approaches is challenging [20]. Furthermore, there remain several drawbacks in the humanized system to overcome, most pertinently, some immune subsets are underrepresented or fail to acquire full functionality, mainly due to a lack of species cross-reactivity and impaired lymph node development. Efforts are underway to characterize the developing immune system and compare the advantages and disadvantages among different immunodeficient host strains [21,22,23,24].

Humanized intraperitoneally and subcutaneously injected ovarian cancer xenograft models have been used in different immunotherapeutic approaches, such as exogenous IL-12 injection, chimeric antigen receptor T cell infusion, and Treg depletion. However, none of those xenograft models have been co-transplanted with a full human immune system, nor have they been xenografted orthotopically [25,26,27,28]. Since many patients experience complete or partial treatment resistance, advanced preclinical humanized models for HGSOC are required to identify (1) new treatment modalities for this treatment resistant group and (2) immunological biomarkers that may predict responders and non-responders to these new therapies. In this study, we developed the very first orthotopic cell-line (CDX) and patient-derived xenograft (PDX) models of HGSOC transplanted with umbilical cord blood-derived CD34^+^ cells. We longitudinally characterized the developing immune system in NSG and NSGS mice and defined the optimal strain with the most suitable variables. The subsequent humanized orthotopic HGSOC models permitted the study of tumor immune cell infiltration and response to anti-PD1 monotherapy in the context of cancer cells, non-malignant tumor-associated cells, and immune cell interaction within the TME.

## 2. Materials and Methods

### 2.1. Generation of Humanized NSG and NSGS Mice

All animal experiments were conducted in compliance with the procedures from the Norwegian State Commission for Laboratory Animals and approved by the Norwegian Food Safety Authority (Application ID 14128). Female *NOD.Cg-Prkdc^scid^ Il2rg^tm1Wj/^SzJ* (NOD-*scid* IL2rγ^null^, NSG) and NOD.Cg*-Prkdc^scid^* Il2r*g^tm1Wjl^* Tg (CMV-IL3, CSF2, KITLG) 1Eav/MloySzJ; (NSG-SGM3; NSGS) mice (aged 8–12 weeks; Vivarium, University of Bergen) were housed in individually ventilated (HEPA-filtered) cages at the University of Bergen’s animal facility. No more than five mice were kept in each cage; food, water, bedding, and cages were autoclaved and changed. Animals were sacrificed according to institutional guidelines at the endpoints defined in the study.

Hematopoietic CD34^+^ cells were isolated from umbilical cord blood, collected from healthy women with presumed healthy pregnancies delivered by cesarean section (Research Biobank for Blood Diseases, HUS, Bergen, Norway). Written informed consent was obtained from all parents before collection. Immediately after the cord of the neonates was clamped, the umbilical vein was punctured with a 21G needle at the distal portion of the cord and 30–60 mL of umbilical cord blood was collected in ACD-A vacutainers (Cat# 01049, BD Biosciences, Franklin Lakes, NJ, USA). For the isolation of mononuclear cells, we diluted cord blood 1:1 with sterile PBS and added up to 20 mL of diluted blood slowly to a 50 mL falcon tube containing 10 mL Lymphoprep^TM^ (Cat# 1114545, Axis Shield Diagnostics Ltd., Dundee, Scotland). The falcon tubes were centrifugated at 400 *g* with low acceleration and without brakes for 25 min at room temperature (RT). After centrifugation, the mononuclear cells, which appeared as an opaque interphase, were collected, washed with PBS, and centrifugated (500 *g*, 5 min, RT). Red blood cells (RBC) were lysed with RBC lysis buffer (155 mM NH_4_CL, 10 mM NaHCO_3_, 0.1 mM EDTA, pH 7.3 in milli-Q water) for eight minutes, protected from light and centrifugated in sterile-filtered magnetic associated cell sorting (MACS) buffer (PBS, 2 mM EDTA, 0.5% bovine serum albumin). Subsequently, hematopoietic CD34^+^ cells were isolated using positive-selection MACS as described by the manufacturers (CD34 MicroBeads Kit cat# 130-046-702, Miltenyi Biotec, Bergisch Gladbach, Germany). The purity of anti-human CD34-PE (cat# 130-098-140, clone: AC136 Miltenyi) stained and enriched CD34^+^ cells was analyzed by flow cytometry. Samples with a purity greater than 95% (95.6 ± 2.1%) were cryopreserved in CryoStor-10 (cat# 07,930 BioLife Solutions Inc, Bothell, WA, USA).

NSG and NSGS mice were pretreated with 25 mg/kg busulfan (or as otherwise indicated) (6 mg/mL, Vnr# 582266, Fresenius Kabi, Bad Homburg von der Höhe, Germany) one day before 1 × 10^5^ CD34^+^ cells (or as otherwise indicated) were intravenously injected via the tail vein.

### 2.2. Generation of Humanized Xenograft Models

Human OV-90^luc+^ cells were obtained from the American Type Culture Collection (ATCC, Manassas, VA, USA). The cell line was cultured in RPMI 1640 media (cat# D5671, Sigma Aldrich, St. Louis, MO, USA) supplemented with 10% heat-inactivated fetal calf serum (FBS, cat# 10270106, Gibco, Paisley, UK) and 1% L-glutamine (cat# 25030081, Gibco). The stable expression of red-shifted *Luciola Italica* luciferase (cat# CLS960003, Perkin Elmer, Waltham, MA, USA) allowed for non-invasive in vivo monitoring of tumor growth by bioluminescence imaging (BLI). CDX xenograft models were generated seven to thirteen weeks after humanization by subcutaneous injection of 5 × 10^6^ or orthotopic injection of 1 × 10^5^ OV-90^luc+^ cells, as described previously [29].

Patient tumor samples were provided by the Gynecologic Cancer Biobank, Women’s Clinic, Haukeland University Hospital, Bergen, Norway (REK ID: 2014/1907, 2015/548, 2018/72). Written informed consent was obtained from all women before collection of fresh tumor tissues and clinico-pathological parameters were initiated. Treatment naïve patient ovarian tumors sampled from primary debulking surgery and tissues from the consequent engrafted PDX model were collected in RPMI medium and processed within one hour. The tumor tissues were cut into small tissue pieces of 1 mm^3^ in size and were enzymatically dissociated with collagenase II (300 U/mL, cat#17101015, Gibco) and DNase (0.1 mg/mL, cat# 07900, STEMCELL Technologies, Cambridge, UK) supplemented with calcium chloride (3 mM) for two hours with constant agitation (250 rpm) at 37 °C. Digested tumors were washed in PBS, strained through a 40 μm cell strainer and centrifugated; cell viability was determined with trypan blue staining. Samples were cryopreserved and immediately injected orthotopically into the bursa of the ovary [29]. After successful engraftment into NSG mice, the passaged PDX material was cryopreserved before further in vivo preclinical studies. Two PDX models have been used in this study. PDX26 is an HGSOC stage IV primary tumor, and PDX19 originates from an ovarian tumor of unknown primary origin (CUP). Due to different engraftment kinetics, the PDX material from both patients were engrafted at different time points. Orthotopic PDX26 xenograft models were generated two weeks after intravenous injection of CD34^+^ cells, whereas 1 × 10^5^ cells isolated from patient 19 were orthotopically injected nine weeks after CD34^+^ cell injection.

### 2.3. Evaluation of Tumor Growth

Caliper measurements of subcutaneous tumors were performed twice weekly ((height × width × length × π)/6). Orthotopic tumor growth was evaluated by optical imaging. BLI was performed using the IVIS Spectrum In Vivo Imaging System (Perkin Elmer) 10 min after intraperitoneal administration of 150 mg/kg of D-luciferin (cat # L-8220, Biosynth, Staad, Switzerland). Tumor progression in animals xenografted with luciferase positive OV-90^luc+^ cell lines was monitored weekly following tumor cell injection and during treatment.

Fluorescence imaging (FLI) scans (λ_ex_ = 670 nm, λ_em_ = 700 nm LP, laser repetition rate 80 MHz, raster scan points 1 mm apart) were acquired on depilated PDX models pre- and post-treatment with the Optix MX3 Small Animal Molecular Imaging system (ART Inc., Saint Laurent, QC, Canada). The monoclonal antibody CD24 (clone SN3, cat # MCA1379, Bio-Rad, Oxfordshire, UK), labeled with Alexa Fluor 750^®^ (DOL = 2.24) (cat# S30046, Thermofisher Scientific, Waltham, MA, USA), was injected intravenously via the tail vein at a concentration of 2 μg/g 24 h before FLI acquisition [29].

### 2.4. Treatment

Busulfan preconditioned (25 mg/kg) NSG mice were intravenously injected with 100,000 CD34^+^ cells. After successful chimerism, defined as more than 25% human CD45^+^ cells in the peripheral mouse blood samples, humanized NSG xenografts were treated with nivolumab (anti-PD-1, 10 mg/mL, OPDIVO, ATC# LO1X C17, Bristol-Myers Squibb, NY, USA) starting at week 12. Ten orthotopic engrafted OV-90^luc+^ mice were allocated into treatment and control arms according to chimerism and to tumor load, measured by BLI three weeks after tumor engraftment. Two established PDX models, those with patient IDs 19 and 26, were engrafted in four mice each and divided into two groups. Tumor growth proceeded for ten weeks in PDX 26 and for 3 weeks in PDX 19. The treatment group received 2 mg/kg twice a week for two weeks followed by two doses of 100 mg/kg of nivolumab intraperitoneally. Tumor growth and therapeutic responses were monitored by BLI (weekly in CDX models) and FLI (pre- and post-treatment in PDX models).

### 2.5. Evaluation of Immune Cell Subsets

Peripheral blood (PB) was drawn longitudinally from the facial vein of each mouse. Whole blood samples were analyzed with flow cytometry, ProCyte IDXX^®^ Hematology Analyzer (IDEXX Laboratories, Westbrook, ME, USA), and mass cytometry. At the endpoint, tumor tissues were processed as previously described for mass cytometry analysis as well as for immunohistochemistry. Blood cell count analysis with the ProCyte DX^®^ Hematology Analyzer was performed with 50 μL of PB collected in EDTA microvettes (cat# 20.1341.100, Sarstedt AG & Co. KG, Nümbrecht, Germany) to characterize the clinical features of anemia (platelets, RBCs, reticulocytes, hemoglobin).

### 2.6. Flow Cytometry

PB (50 μL) was collected from the facial vein in EDTA microvettes. Non-specific Fc receptor-mediated interactions were blocked with 10% human Fc receptor block (cat# 130-059-901, Miltenyi) for 10 min at RT. Followed blocking, samples were stained for 15 min at 4 °C in a staining volume of 100 µL, protected from light. Fluorochrome-conjugated mAb against the following antigens were used: human CD45-PerCP-Vio700 (clone 5B1, cat# 130-097-527, Miltenyi), mouse CD45-FITC (clone 30F11, cat# 130-102-491, Miltenyi), human CD19-BV421 (clone HIB19, cat# 562440, BD Biosciences), and human CD3-APC-Cy7 (clone SK7, cat# 560176, BD Biosciences). Prior to washing, red blood cells were lysed with 1 mL of RBC lysis buffer (155 mM NH_4_CL, 10 mM NaHCO_3_, 0.1 mM EDTA, pH 7.3 in milli-Q water) for five minutes at RT. Once the lysis was completed, samples were centrifugated and washed twice with FACS staining buffer (PBS pH 7.4 supplemented with 5% FBS and 2 mM EDTA). Pellets were resuspended in FACS staining buffer supplemented with 1 μg/mL propidium iodide (cat# P48640, Sigma Aldrich) before data acquisition with a BD Fortessa Flow Cytometer using FACSDIVA software (both BD Bioscience, Franklin Lakes, NJ, USA).

### 2.7. Mass Cytometry

Tumor single cell suspensions were resuspended in Maxpar PBS (cat# 201058, Fluidigm, San Francisco, CA, USA) supplemented with 5 mM EDTA, and stained with Cell-ID Cisplatin (cat#201064, Fluidigm) for 60 s at a concentration of 5 μM to identify dead cells. The reaction was quenched with Maxpar Cell Staining Buffer (CSB) (cat# 201068, Fluidigm), samples were centrifugated, and the pellet resuspended in CSB. Blood and tumor samples were lysed and fixed in stable-lyse stable-store V2 buffer (cat# STBLSTORE2-1000, STBLYSE-250, Smart-Tube Inc., San Carlos, CA, USA) and stored at −80 °C according to the manufacturer’s recommendations. Frozen, fixed, and cisplatin-stained single-cell suspensions of xenografted tumor tissue and blood were thawed, washed twice in CSB, and prepared for palladium barcoding (Cell-ID 20-Plex Pd Barcoding Kit, cat# 201060, Fluidigm). For tumor barcodes, 3 × 10^6^ cells were aliquoted into individual tubes and permeabilized with 1× Barcode Perm Buffer. Samples were barcoded for 30 min as described previously by Zunder et al. [30]. For the humanized blood barcode, 100–200 μL fixed blood was used from pre-treatment samples (week 11) and post-treatment samples (survival endpoint). The pooled, barcoded samples were washed twice in CSB and resuspended in CSB supplemented with 10% Fc receptor-blocking solution (cat# 130-059-901, Miltenyi) and 200 U/mL heparin and incubated for ten minutes at RT. Without washing, cells were stained for 30 min at RT with the surface antibody cocktail (Appendix A) in a staining volume of 100 μL per 3 × 10^6^ barcoded cells. Following incubation, samples were washed twice with CSB and the DNA was stained with Cell-ID^TM^ Intercalator-Iridium solution (1:10 Maxpar Permeabilization Buffer; 1:4 16% PFA; 1:8000 (blood) or 1:16,000 (tumor) Iridium (cat# 201192A, Fluidigm); in Maxpar PBS) overnight at 4 °C to allow the identification of single cells [31]. Before sample acquisition, cells were washed in CSB, followed by PBS, and resuspended in CAS solution (cat# 201237, Fluidigm) supplemented with normalization beads (1:10). The WB injector in the sample introduction system of the Helios^TM^ mass cytometer was used for data acquisition with Helios 6.5.358 acquisition software (Fluidigm).

### 2.8. Immunohistochemistry

Primary tumor samples from xenografted hu mice were fixed in 4% paraformaldehyde and embedded in paraffin. Formalin fixed paraffin embedded (FFPE) samples were sectioned and prepared for immunohistochemical staining with rabbit anti-human antibodies against CD45 (polyclonal, 1:2000, cat#ab10558, Abcam) and CD3 (clone SP7, 1:200, cat# RM-9107-S, Thermofisher Scientific, Waltham, MA, USA). Slides were deparaffinized and antigen retrieval was performed using LpH buffer for CD45 and HpH buffer for CD3 using Dako PT Link. Dako EnVision was used for immunostaining, using rabbit-specific polymer-HRP to avoid false-positive mouse-mouse cross reactivity in the mouse tissue. Human tonsil tissue was used as positive control.

### 2.9. Data Analysis

Barcoded files were normalized against EQ^TM^ Four Element Calibration Beads (cat# 201078, Fluidigm) with the Fluidigm software and debarcoded (GitHub premessa debarcoding R package) into up to 20 individual files, each representing one sample. All samples were cleaned up by manual gating in Cytobank. Live cells and singlets were selected through gating and beads and cells of murine origin were removed before the files were introduced into dedicated downstream analysis tools. Samples were analyzed by phenotypic spanning-tree progression analysis of density-normalized events (SPADE), cell clustering and dimensional reduction with preservation of single-cell resolution with the viSNE algorithm in Cytobank (t-distributed stochastic neighbor embedding, t-SNE) [32,33]. Different immune populations were identified by manual gating on calculated dimensionality-reduced viSNE maps embedded in the SightOf MATLAB tool [32].

### 2.10. Statistical Analyses

Statistical analyses were performed using GraphPad Prism software (Version 6.0, La Jolla, CA, USA). To test for significant differences between two variables, we used a two-tailed nonparametric unpaired Mann–Whitney U test where *p*-values < 0.05 were regarded as statistically significant. All data with error bars are presented as mean ± standard deviation. Survival curves for treated and untreated humanized OV-90^luc+^ CDX, PDX19 and PDX26 models were plotted using Kaplan–Meier and the significance between both groups was tested with the Mantel–Cox log-rank test. A Wilcoxon matched-pairs signed rank test was performed to assess the difference between NSG and NSGS T cell reconstitution.

## 3. Results

### 3.1. Parameters Influencing Human Hematopoiesis of Humanized NSG and NSGS Mice

To improve the generation of reproducible humanized (hu) mice, we addressed the effect of three different parameters that are known to influence human hematopoiesis. Those parameters were the host mouse strain, bone marrow preconditioning with busulfan and the number of injected CD34^+^ cells. Eight weeks after injection of CD34^+^ cells, blood samples from preconditioned and non-preconditioned hu NSG and NSGS mice were drawn regularly (biweekly or every 4 weeks) and analyzed by flow cytometry (Figure 1a,b).

First, we evaluated the chimerism level in NSG and NSGS mice over time. Robust engraftment (defined as >25% human CD45^+^ hematopoietic cells in mouse blood) in hu NSG and hu NSGS mice was observed after eight weeks (NSG, 48.5 ± 21%, *n* = 26; NSGS, 41.6 ± 19%, *n* = 15) and remained stable with little deviation over time (NSG, 46.5 ± 21.5%; NSGS, 34.0 ± 20.5%) (Figure 1c). Statistical analysis detected no significant difference in the chimerism level between the two strains (*p* > 0.05). The percentage of B cells (CD45^+^CD19^+^) declined after initial engraftment, whereas T cell development occurred ten weeks after engraftment, independent of the mouse strain (Figure 1d). The frequency of T cells in NSG mice increased until week 24, whereas it stabilized after 16 weeks in NSGS mice (Appendix A).

NSGS mice developed severe anemia between 8 and 16 weeks after CD34^+^ cell transplantation, as observed by pallor, weight loss and low bone marrow cellularity. Therefore, we next assessed the effect of busulfan bone marrow conditioning by testing different busulfan concentrations (Table 1). The measurements of red blood cells, hemoglobin, and platelet numbers in the peripheral blood revealed a decrease in all values in both low and high busulfan-dosed animals as well as in non-conditioned hu NSGS mice after CD34^+^ cell injection over time. The decline was, however, greatest in NSGS mice treated with the standard dose of 25 mg/kg busulfan (Table 1), suggesting that the severity of anemia can be delayed and diminished with decreased busulfan concentration. NSGS mice preconditioned with high-dose busulfan showed the highest levels of chimerism and CD45^+^CD3^+^ cell frequency among the three groups until 12 weeks after humanization (Figure 1e,f). The effect of the high busulfan concentration on improved chimerism and increased frequency of T cells disappeared after 16 weeks.

The biggest variation in chimerism levels, as well as in lymphocyte populations, was observed between different cord blood donors with small inter-mouse variation within one experiment (Figure 1d, Appendix A). To improve the experimental design by injecting a larger cohort with CD34^+^ cells from the same cord blood donor, we evaluated the influence of the number of injected CD34^+^ cells on chimerism level and differentiation into human T cells. The comparison between 0.5 × 10^5^ and 1 × 10^5^ injected CD34^+^ cells revealed no impact on the percentage of human hematopoietic cells in NSG blood (*p* > 0.05) (Figure 1e), but a larger number of CD34^+^ cells modestly favored the development of CD45^+^CD3^+^ cells in the mice (Figure 1f). However, this effect was not significant. When comparing the frequency of T lymphocytes at different time points between NSG and NSGS hu mice, we found significantly elevated levels of T cells in NSGS mice constitutively expressing human IL-3, GM-CSF, and SCF (*p* = 0.0078) (Figure 1d,f, Appendix A).

### 3.2. Characterization of Human Hematopoietic Cells in Peripheral Blood of Hu Mice

To address the capacity of NSG and NSGS mice to support hematopoiesis from CD34^+^ cells into the different cell subsets of the myeloid and lymphoid human immune system, we stained peripheral blood samples (16–24 weeks after human immune reconstitution) with a panel of 34 surface markers for mass cytometry (Appendix A). Manual gating was performed on dimensionality reduced viSNE maps of equal numbers of human CD45^+^ hematopoietic cells (15,000 cells) from NSG and NSGS mice, resulting in eight major cell populations. Between the two strains, we identified several discrepancies in the frequencies of cell populations in the peripheral blood (Figure 2a).

Complete human T cell education in the murine thymic and lymph node environment is of principle importance to successfully evaluate the effect of ICI. We characterized the T cells (Figure 2b, dashed circle) by surface staining with nine differentiation and activation markers and identified CD4 and CD8 T lymphocytes that could be subdivided into Tregs, naïve T cells, effector memory (TEM) and central memory (TCM) T cells (Figure 2b and Figure 3a). Most of the T cells expressed CD45RO and were therefore attributed to differentiated memory T cells (Figure 2b). In addition, we found NK cells, monocytes, dendritic cell (DC) subsets (CD1c^+^, CD141^+^ and CD11c^+^ mDCs), B cells and CD45^low^ cells in both strains (Figure 3a,b, Appendix A). NSG mice favored the development of B cells, whereas many more T cells (as previously shown by flow cytometry, Figure 1f, Appendix A), especially regulatory and activated subsets, and myeloid-derived cells were observed in the peripheral blood of NSGS mice (Figure 3b). In addition, different NK cells subsets were identified with greater frequency in NSG mice (Figure 3b, Appendix A). Most NK cells were CD335^+^ with a CD56bright CD62Lint phenotype, whereas a smaller population of CD335^+^ CD56^dim^ CD16^+^ cytotoxic NK cells and CD335^+^CD56^dim^CD16^−^CD127^+^ precursor NK cells were present (Supplementary Figure 2b).

### 3.3. Subcutaneous and Orthotopic Tumor Growth Was Not Impaired in Humanized Mice

We aimed to subcutaneously and orthotopically engraft established hu mice with an OV-90^luc+^ cell line to generate relevant hu CDX mice with HGSOC to further test immunotherapies. Hu mice were subcutaneously xenografted with 5 × 10^6^ OV-90^luc+^ cells into the left and right flank (*n* = 12) 7 to 13 weeks after CD34^+^ cell injection. The tumor growth kinetics were compared to immunodeficient NSG and NSGS mice (*n* = 6). Successful engraftment was observed at a frequency of 100%, and tumor growth was comparable between hu OV-90^luc+^ and OV-90^luc+^ mice (Figure 4a). A reconstituted human immune system also had no influence on tumor growth, tumor dissemination, or the survival of orthotopic implanted OV-90^luc+^ hu mice (*n* = 8, *p* > 0.05) (Figure 4b). In addition to CDX models, we generated two humanized orthotopic PDX models of ovarian cancer (Figure 5). Our results indicated that hu mice allowed the growth of both ovarian cancer cell lines and patient-derived tumor samples.

### 3.4. PD-1 Blockade Inhibited Tumor Growth in Hu OV-90 Xenograft and Hu PDX Models

Advanced ovarian cancer remains a challenging disease to treat. ICI have on overall reported response rate of 6–22% in HGSOC [34]. Regardless, ICI is employed to boost the anti-tumor T cell response in the ovarian TME, which in combination with other treatments might lead to efficient disease control [34]. Therefore, we utilized the generated hu HGSOC xenograft models to study the response to anti-PD-1 immunotherapy. Two hu orthotopic PDXs (PDX26, *n* = 2; PDX19, *n* = 2) and hu OV-90^luc+^ CDXs (*n* = 5) were treated with an anti-PD-1 (nivolumab) ICI, an approach that has already been evaluated in late ovarian cancer clinical trials. Survival, fluorescence and bioluminescence intensities, and immune cell composition of the treated mice were compared to an equal number of CDX and PDX controls. The experimental design and treatment plan are outlined in Figure 5a.

We previously demonstrated that reconstituted activated T cells express PD-1 (Appendix A). We evaluated chimerism and the frequency of T cells in the blood pre and post-nivolumab treatment (Figure 5b). Mice were allocated to treatment and control arms based on 11-week chimerism and T cell levels. All mice responded to nivolumab with progressive disease. However, survival in all three models was prolonged (control, 54.4 ± 20.2 days; treated, 61.9 ± 26.0 days) (Figure 5c). BLI of nivolumab treated OV-90^luc+^ hu mice (*n* = 5) indicated a decrease in tumor burden compared to the untreated control cohort (*n* = 5) (Figure 5d, Appendix A) and prolonged survival by 5 days (disease latency 43 ± 4). Two non-hu OV-90^luc+^ mice treated with nivolumab (*n* = 2) showed comparable disease latencies to hu OV-90^luc+^ untreated control mice, indicating that the treatment effect may be dependent on a functional human immune system (Appendix A). The benefit of anti-PD-1 targeted immunotherapy was also observed in hu PDX mice as a reduction in fluorescence signal intensity (Figure 5d,e). At the humane endpoint, both PDX models showed disseminated disease to the omentum, liver, diaphragm, mesentery, spleen and ascites, in both treatment and control arm (Appendix A). Survival was prolonged by 17 days in PDX26 (disease latency 96 ± 11 days) and 6 days in PDX19 (disease latency 59 ± 9 days) but this was not statistically significant (Figure 5c). PD-L1 expression level is one of the leading biomarkers for predicting the response to ICI for various cancer types and is evaluated for stratification of EOC patients [9,35]. Therefore, we evaluated the correlation of PD-L1 expression on tumor cells with the response to nivolumab treatment. Tumor cells derived from PDX19 xenografts had increased PD-L1 expression on more than 75% of analyzed tumor cells (Figure 5f). The high expression of PD-L1 was confirmed for the original patient tumor (Figure 5f). In contrast, only around 0.7% of OV-90 and 2% of PDX26 tumor cells were positive for PD-L1 staining. High PD-L1 expression in hu PDX19 did not improve the efficacy of treatment, and in all three hu xenograft models, we observed no influence of PD-L1 expression on survival.

### 3.5. Characterization of Tumor-Infiltrating Immune Cells after Nivolumab Treatment

To assess immune modulation in response to ICI, we characterized the phenotypic composition of the immune cells in the blood and infiltrating the TME before and after treatment with a panel of 34 surface markers and mass cytometry. First, we identified a significant decrease in T cells expressing PD-1 in the peripheral blood after nivolumab treatment (*n* = 9); such a decrease was not observed in the control cohort (*n* = 8) (Figure 6a). When we assessed the tumor-infiltrating immune cells in the control and treated cohort, we identified a significant decrease in PD-1 expression for the entire immune cell population (CD45^+^ cells) in nivolumab-treated animals (Figure 6b). We then asked whether anti-PD-1 treatment affected tumor immunosuppression by measuring immune cell subset infiltration of the ovarian TME. The tumors in treated PDX xenografts showed increased infiltration by NK cells (Figure 6c). In addition, the number of tumor-infiltrating CD8 T cells did increase in the immune “hot” PDX26 model, a hallmark for the response to anti-PD-1 therapy [36]. Lastly, most tumor-residing immune cells belonged to the myeloid lineage. However, the generally low number of human tumor-infiltrating immune cells limits in-depth phenotypic characterization of these cells. The infiltration of CD45^+^ cells was additionally confirmed by IHC (Appendix A). No differences between control versus treated xenograft tumors were observed. We confirmed that PD-1 is expressed on TEM and TCM in the blood of hu mice and assessed the effect of nivolumab treatment on T cells in the peripheral blood (Figure 6d). We could not identify any significant changes in the frequencies of human immune cell subsets in control and treated PDX blood samples (Figure 6d). Blood samples taken before treatment indicated differentiated T cells and an increased CD8:CD4 ratio compared to post-treatment. The composition of the immune system in OV-90^luc+^ hu mice changed over time and between control and treated animals (Appendix A). Specifically, the frequency of CD4 and CD8 T cells increased after anti-PD-1 treatment in the blood of OV-90^luc+^ hu mice.

## 4. Discussion

Advances in targeted immune therapy for many solid tumors has greatly benefited patients. However, only a minority of patients are responders, and the reasons for this remains still largely unknown. Patient stratification is only possible if the immunological features that lead to an improved outcome, with all its complexity and underlying pathways of resistance, are understood. To improve clinical translation of immune modulating drugs for oncology patients, advanced preclinical mouse models must harbor a human immune system together with heterogeneous tumors xenografted in a physiological microenvironment. In this study, we successfully developed orthotopic humanized CDX and PDX models that provided insights into the individual response to immunotherapy.

We first characterized the immune system generated in both NSG and NSGS mouse strains with the use of a panel of 4 surface markers and flow cytometry, and a panel of 34 surface markers and mass cytometry, which allowed us to identify various lymphoid and myeloid cell types. Whereas NSG mice favored the development of B cells, increased numbers of T cells, especially regulatory and activated T cells, as well as myeloid-derived cells were observed in the NSGS strain. Differentiation and maturation of the majority of T cells into CD45RO-expressing effector memory and central memory T cells showed that allogeneic CD34^+^-derived progenitor cells underwent positive selection in the mouse thymus, which is pivotal for the generation of tumor-specific T cells [37]. For effective anti-tumor cytotoxicity and maintenance of peripheral tolerance, cross-presentation of neoantigens by tumor myeloid antigen presenting cells is essential [7,24]. We identified a variety of DC subsets in both strains, with the largest being CD11c^+^ DCs, followed by CD141^+^ mDCs and a minor subset of CD1c^+^ mDCs. Improvements in hu mouse development that favor physiological cytokine levels, cross-presentation and lymphoid environments, such as synthetic lymphoid-like organoids, are ongoing. Knock-in strains, such as MISTRG, MITRG and NSG-HLA, and co-transplantation of human thymus in BLT mice may address the aforementioned limitations [38,39,40]. However, these advanced knock-in strains have a shorter lifespan compared to the NSG and NSGS strains, rendering them unsuitable for the generation of humanized PDX models with an engraftment time of 12 weeks or longer (Figure 5a) [18].

Currently, there is no consensus on the parameters essential for creating reliable, stable chimerism in hu mice. Specifically, donor-to-donor variation in chimerism and immune composition affects the design of treatment studies. Most humanized mice are generated by intrahepatic injection of irradiated newborn mice with 1 × 10^5^ CD34^+^ hematopoietic progenitor cells [18,21,24,41]. By using intravenous injection and 8- to 12-week-old NSG and NSGS mice, we showed that the number of injected stem cells could be reduced without affecting engraftment. In addition, we observed precondition-dependent anemia in NSGS mice, most likely the result of stem cell exhaustion, which was also indicated by oscillating chimerism levels [18,42]. We showed that decreased erythropoiesis could be overcome by avoiding toxic preconditioning with busulfan. A great deal of effort has been made to improve human erythro-megakaryopoiesis in mice with a loss-of-function KIT (CD117) receptor (NSGW41), overexpression of human erythropoietin, depletion of mouse macrophages and thrombopoietin knock-in (MISTRG), which all showed improved myelopoiesis, erythropoiesis, and platelet formation [43,44,45].

We are the first to show the generation of orthotopic CDX and PDX hu mice co-engrafted with a human immune system and HGSOC disease, which represents a promising model to evaluate the efficacy of ICI, as shown for other solid tumors [46,47]. Bankert et al. showed that autologous tumor-infiltrating immune cells remained functional after intraperitoneal co-injection of tumor cells and tumor stroma obtained from OC patients, [25]. Dual ICI with anti-CTLA-4 and anti-PD-1 was tested in a similar autologous PBMC intraperitoneal EOC PDX model with a significant reduction in tumor volume [28]. This approach enables the study of a patient’s immune response with matched immune-tumor cell donors shortly after TIL or peripheral blood mononuclear cells (PBMC) injection, avoiding the long reconstitution time after CD34^+^ hematopoietic progenitor cell injection. However, unstimulated PBMCs and infiltrating tumor lymphocytes die after a few weeks [48]. Furthermore, NSG mice engrafted with human PBMCs may develop severe xenogeneic graft-versus-host disease (GVHD) beginning 3-4 weeks after injection [49]. Hence, isolation of CD34^+^ cells from the patient’s bone marrow would overcome this limitation. Our model incorporates, in addition to the patient's tumor and stromal cells, a complete humanized immune system, which can be educated to infiltrate the TME and attack a tumor that has been established orthotopically in the ovary.

Mice treated with nivolumab had progressive disease with lower tumor burden and non-significant, but prolonged survival compared to untreated control mice. Nivolumab releases the brake on the inhibitory costimulatory signal between antigen-presenting cells and T cells, leading to the expansion of specific tumor-infiltrating cytotoxic CD8 T cells [28,50]. Therefore, the response to PD-1 ICI requires an immunogenic tumor (so-called “hot tumors”) so that existing T cells can be reinvigorated [12]. Profiling of harvested tumors showed the presence of human immune cells. Hot tumors with increased tumor infiltration by CD4 and CD8 T cells were observed in treated hu PDX26 mice, but not in hu PDX19 mice that expressed high levels of PD-L1. A PD-L1 expression level of ≥ 1% of tumor cells is one of the leading immune-related biomarkers that predicts a response to ICI for various cancer types. Similar to data from two EOC trials, we could not identify any correlation between PD-L1 expression, response rate and survival [8,9]. Generally, the treatment efficacy in treated PDX and OV-90^luc+^ mice was low and comparable with the overall response rate of 15% in EOC patients treated with nivolumab monotherapy [51]. Tumor burden, immunosuppressive TME, and exhausted T cells are negative prognostic factors for the response to ICI [52,53]. In our model, tumor burden was high with already-metastasized OV-90^luc+^ xenografts in the mice at treatment start. ICI has been suggested for patients responding to first-line treatment with a decreased tumor burden to allow immune stimulation and education. Therefore, the response to ICI as indicated by tumor shrinkage might occur later following initial progression [54]. Regarding immunosuppressive TME, we identified a high frequency of myeloid cells, which supported myeloid differentiation in the NSG strain but may also indicate an immunosuppressive TME [47,55] that is correlated with a poor prognosis [56]. The TME in EOC is a complex and dynamic interactive entity, with multiple immunosuppressive mechanisms. One of the first components to be defined for the immunosuppressive TME in EOC was Treg-mediated immunosuppression of tumor-specific T cells, which was associated with reduced survival [57]. The immunosuppressive myeloid compartment comprises tumor-associated macrophages (TAMs), including CD163^+^CD204^+^ tumor associated macrophages (M2), MDSC (myeloid derived suppressor cells), immature mDCs and pDCs. These subsets have been correlated with resistance to immune therapy, increased tumor burden and early relapse in EOC patients [58,59]. In contrast, antigen-presenting myeloid cells activate tumor-specific T cells and support their effector fitness [7]. However, due to low numbers of CD45^+^ cells acquired for mass cytometry analysis, in-depth phenotypic characterization of the myeloid cells was not possible. The third factor associated for a poor response to ICI, exhausted T cells, may have also been present in our experiments, since high PD-1 expression on T cells and tumor infiltrating immune cells might indicate an exhausted state. After ICI, the frequency of T cells expressing PD-1 decreased in the blood as well as in TILs, which was also reported in two other hu mouse models after anti-PD-1 treatment [46,47].

## 5. Conclusions

In conclusion, we generated humanized PDX and CDX models and used deep tissue profiling of immune subsets to evaluate PD-1 monotherapy for HGSOC. The overall response in our monotherapy treatment study was modest, which suggests that combination approaches are needed to broaden the indication for treatment and potentiate the effect of ICI therapy for HGSOC. ICI in combination with drugs targeting the hallmarks of EOC, such as histone deacetylases (HDAC) or anti-angiogenesis inhibitors, has been shown to increase EOC tumor immunogenicity and the overall response rate to 25–30% [47,60,61]. We believe that orthotopic PDX hu models with improved immune reconstitution show potential for investigating rational synergistic combination treatments for HGSOC and identifying predictive biomarkers for stratification of patients based on molecular profiling [62,63].

## Figures and Tables

**Figure 1 cancers-14-03092-f001:**
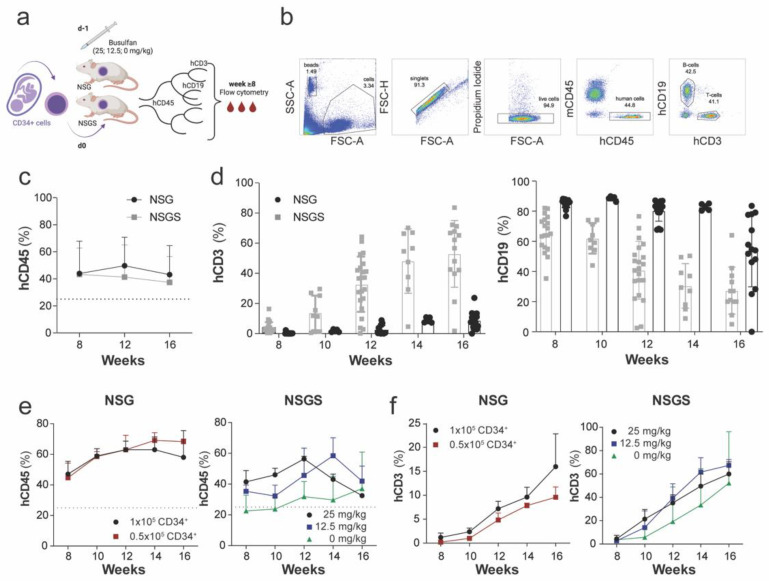
Reconstitution of the peripheral human immune system in NSG and NSGS mice. (**a**) Experimental design: Humanized NSG and NSGS mice were generated by preconditioning with different concentrations of busulfan at day –1, then 0.5–1 × 10^5^ CD34^+^ cells isolated from the umbilical cord were injected intravenously on day 0. Blood samples were drawn regularly between week 8–16 post-engraftment. (**b**) Gating strategy used to define the level of chimerism and T cell development in mouse blood by flow cytometry. (**c**) Longitudinal measurement of human leukocyte frequency in NSG (*n* = 26) and NSGS (*n* = 26) mice. (**d**) Composite data of T cell (left) and B cell (right plot) development in representative NSGS mice (*n* = 25) and NSG (*n* = 15) over time. (**e**) Human leukocyte frequency was compared in NSG mice engrafted with 0.5 × 10^5^ (*n* = 3) and 1 × 10^5^ (*n* = 3) CD34^+^ cells and between different busulfan concentrations in NSGS mice (*n* = 3/group). (**f**) Within the human CD45^+^ leukocyte population, the frequency of CD45^+^CD3^+^ lymphocytes was determined.

**Figure 2 cancers-14-03092-f002:**
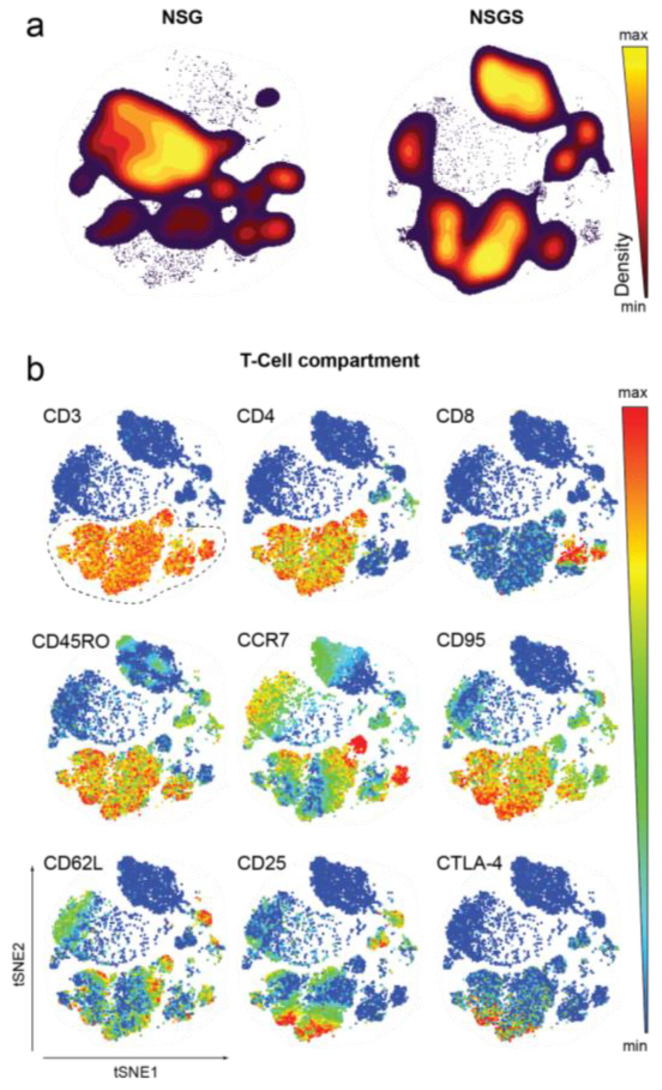
Characterization of peripheral T cells in NSG and NSGS mice with mass cytometry. (**a**) Contour density plots of viSNE maps showing the frequency of human immune cell populations in NSG (*n* = 3) and NSGS (*n* = 3) peripheral blood. (**b**) Phenotypic profiling of the T cell compartment shown for a representative NSGS recipient. The viSNE dot plots are colored according to T cell lineage and activation marker intensity. All marker expression levels are represented from high (red) to low (blue).

**Figure 3 cancers-14-03092-f003:**
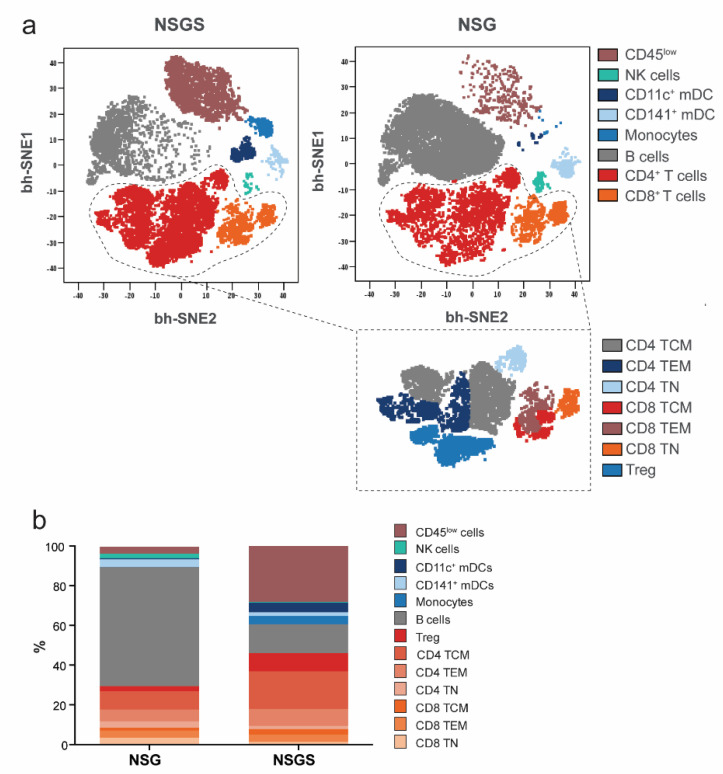
Characterization of peripheral human immune cell subsets in NSG and NSGS mice with mass cytometry. (**a**) Overlay of manually gated cell clusters on viSNE maps. (**b**) Comparisons of all immune cell subset frequencies from NSG and NSGS humanized mice (*n* = 3/each). T cell differentiation defined as: naïve (TN), CD45RO^−^ CD95^−^ CCR7^+^ CD62L^+^; central memory (TCM), CD45RO^+^ CD95^+^ CCR7^+^ CD62L^+^; effector memory (TEM), CD45RO^+^ CD95^+^ CCR7^−^ CD62L^−^; regulatory T cell (Treg), CD4^+^ CD127^−^ CD25^+^ CTLA-4^+^.

**Figure 4 cancers-14-03092-f004:**
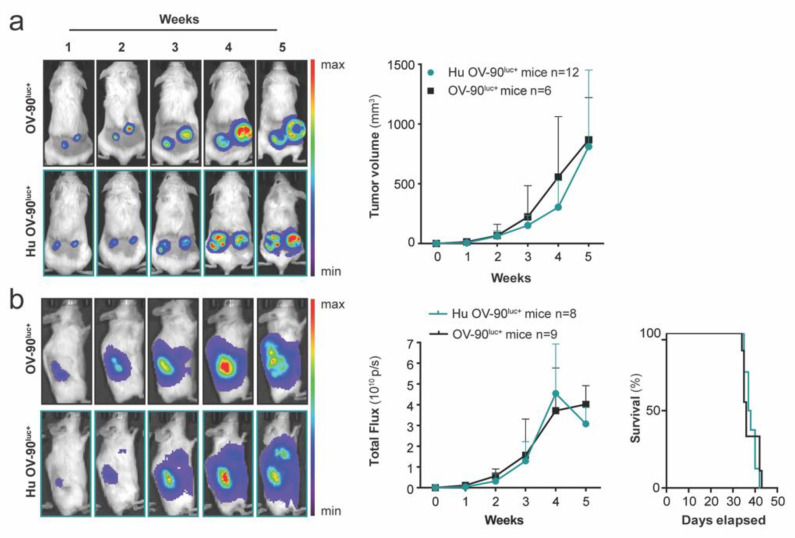
Humanized mice support the engraftment of subcutaneous and orthotopic epithelial ovarian cancer cells (**a**) Humanized (hu) and immunodeficient mice were engrafted subcutaneously with 5 × 10^6^ OV-90^luc+^ cells (*n* =12, hu OV-90^luc+^ mice; *n* = 6, OV-90^luc+^ mice). Tumor growth was monitored by in vivo bioluminescence imaging and caliper measurements ((height × width × length × П)/6). (**b**) Orthotopic OV-90^luc+^ xenografts (*n* = 8, hu OV-90^luc+^ mice; *n* = 9 OV-90^luc+^ mice) were imaged once a week and total flux was calculated. Survival of orthotopic xenografts is presented as days after implantation of 1 × 10^5^ OV-90^luc+^ tumor cells. All optical images are presented as the minimum-to-maximum bioluminescence intensity for each mouse.

**Figure 5 cancers-14-03092-f005:**
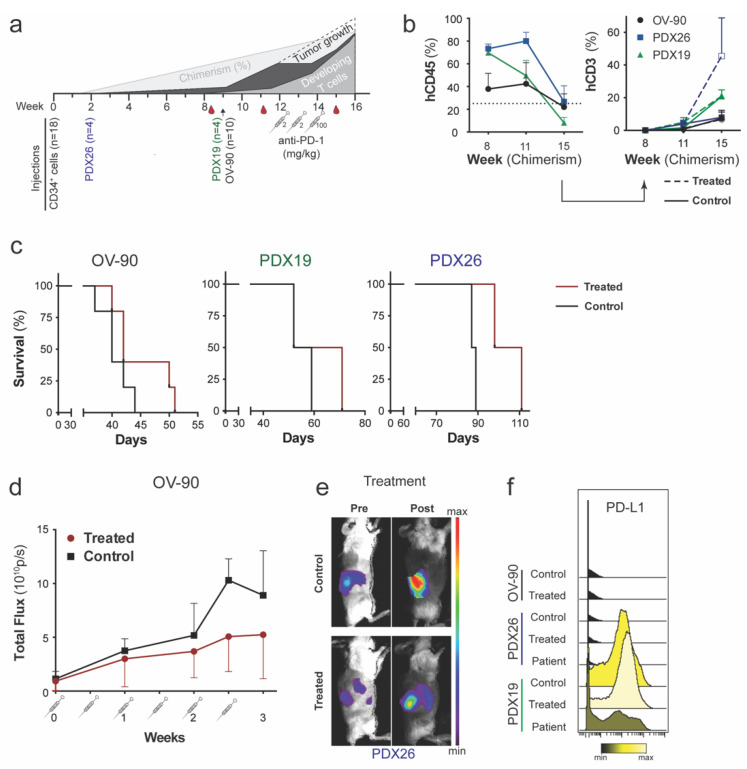
Effect of Nivolumab on tumor growth and prolonged disease latency for an orthotopic OV-90^luc+^ cell line model and two patient-derived xenograft models. (**a**) Experimental design for anti-PD-1 nivolumab treatment with a schematic of expected tumor growth and developing T cells over time. Treatment started 12 weeks after intravenous injection of 1 × 10^5^ CD34^+^ cells in both CDX and PDX NSG models. Mice (*n* = 18) were allocated to treatment and control arms at week 11 according to bioluminescence signal intensity, chimerism level (blood sample week 11) and frequency of T cells (blood sample week 11). The treatment group received 2 mg/kg nivolumab intraperitoneally twice a week for two consecutive weeks and 100 mg/kg twice in the third week of treatment. Blood samples were collected in week 8 to confirm the successful immune reconstitution, in week 11 to allocate the mice into control and treatment groups, and week 15 to assess the effect of treatment. The 11-week samples additionally served as reference pretreatment samples for mass cytometry analysis. (**b**) Frequency of human leukocytes and T cells in mouse blood measured by flow cytometry in weeks 8, 11 and 15 after treatment (weeks from CD34^+^ cell injection). Human leukocytes are shown for all mice (*n* = 18) and CD3 positive cells are presented separately for treated (dashed lines, *n* = 9) and untreated (solid lines, *n* = 9) mice. (**c**) Survival of OV-90^luc+^ (*n* = 10), PDX19 (*n* = 4), and PDX26 (*n* = 4) mice measured as days after the start of tumor engraftment. (**d**) Bioluminescence imaging of OV-90^luc+^ mice was performed to monitor tumor growth during treatment (*n* = 5/group). (**e**) One representative fluorescence image of patient-derived xenografts (PDX26) in control and treated mice pre- and post-treatment. (**f**) Histograms of PD-L1 expression on tumor cells (black = minimum, yellow = maximum median dual count) by mass cytometry.

**Figure 6 cancers-14-03092-f006:**
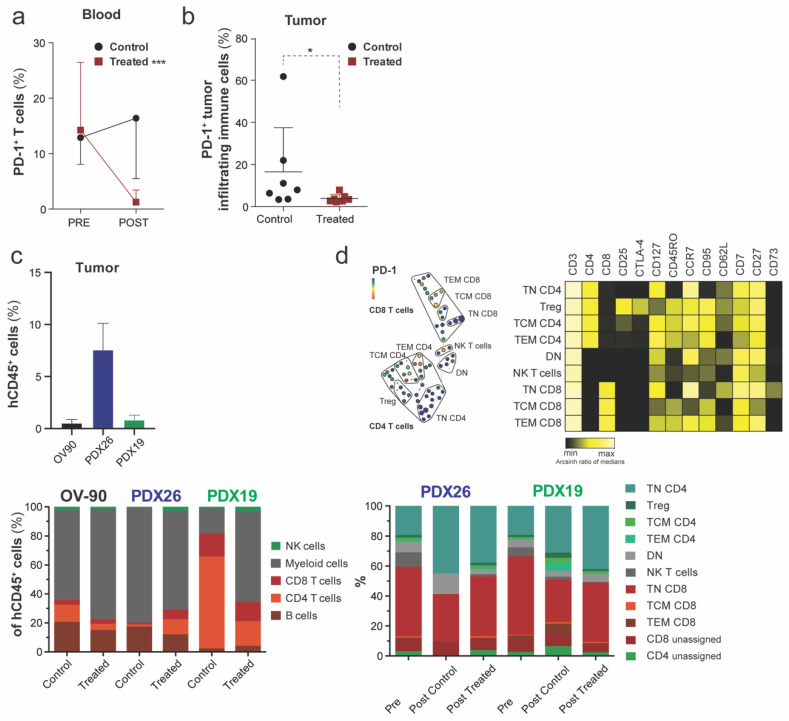
Nivolumab treatment decreases the frequency of PD-1^+^ T cells in the blood and the number of PD-1-expressing tumor-infiltrating immune cells. (**a**) Blood samples collected pre- and post-treatment from treated (*n* = 9, *** *p* < 0.001) and control (*n* = 8) NSG mice were analyzed for T cells expressing PD-1 (% of T cells) by mass cytometry (*n* = 17/18). (**b**) The frequency of PD-1-expressing tumor-infiltrating immune cells in control (*n* = 7) and anti-PD-1 treated (*n* = 7) mice was determined by mass cytometry (* *p* < 0.05). (**c**) Frequency of human CD45^+^ hematopoietic cells in xenograft tumors (upper panel) and the composition of tumor-infiltrating immune cell subsets in control (week 11) and nivolumab-treated OV-90 (survival endpoint, week 14.5–16), PDX19 (survival endpoint, week 16.5–19) and PDX26 (survival endpoint, week 15–18) mice (lower panel). (**d**) Upper panel: SPADE tree clustering of T cells. One representative SPADE tree for a control PDX19 post-treatment blood sample with T cell populations colored according to PD-1 expression in a heatmap (black = minimum, yellow = maximum Arcsinth ratio of medians). Lower panel: frequencies of T cell populations in PDX blood samples from control and treated mice before (Pre, week 11) and after treatment (Post, survival endpoint). T cell differentiation defined as: naïve (TN), CD45RO^−^ CD95^−^ CCR7^+^ CD62L^+^; central memory (TCM), CD45RO^+^ CD95^+^ CCR7^+^ CD62L^+^; effector memory (TEM), CD45RO^+^ CD95^+^ CCR7^−^ CD62L^−^; regulatory T cell (Treg), CD4^+^ CD127^−^ CD25^+^ CTLA-4^+^; double negative (DN), CD4^−^ CD8^−^; natural killer T cells (NK T cells), CD56^+^ CD335^+^ CD14^+^. To test for a significant difference between two variables, we used a two-tailed nonparametric unpaired Mann–Whitney U t-test where *p*-values <0.05 were regarded as statistically significant (* *p* ≤ 0.05; *** *p* ≤ 0.001).

**Table 1 cancers-14-03092-t001:** NSGS mice engrafted with human umbilical cord blood develop pancytopenia.

	NSGS	NSG		NSGS Hu			NSG Hu
			Week	Busulfan Concentration (mg/kg)			
				0	12.5	25	25
**RBC** (M/μL)	8.0 ± 0.32		10	6.28 ± 0.12	4.75 ± 1.78	4.70 ± 0.71	8.12 ± 0.30
		8.8 ± 0.54	14	4.74 ± 0.81	3.99 ± 0.53	3.84 ± 0.76	7.22 ± 0.34
			18	3.04 ± 1.98	2.31 ± 0.38	3.45 ± 0.06	5.92 ± 0.99
**HGB** (g/dl)	13.48 ± 0.51		10	12.1 ± 0.4	9.30 ± 3.38	9.67 ± 1.12	13.50 ± 0.5
		13.96 ± 0.94	14	9.83 ± 1.55	8.50 ± 0.82	8.3 ± 1.01	12.72 ± 0.63
			18	6.73 ± 4.05	5.40 ± 0.99	7.75 ± 0.21	10.74 ± 1.85
**Platelets** (K/μL)	1007.4 ± 69.2		10	635.3 ± 41.2	406.3 ± 174.2	296 ± 103.5	508.3 ± 52.8
		1093.4 ± 94.6	14	550.0 ± 225.4	280.0 ±1 33.4	315.0 ± 109.3	619.0 ± 104.7
			18	228.0 ± 143.4	174.0 ± 144.3	222.0 ± 18.4	544.2 ± 114.0

RBC, red blood cells; HGB, hemoglobin.

## Data Availability

Data available in a publicly accessible repository. The data presented in this study are openly available in FigShare at [10.6084/m9.figshare.19733821].

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
