# Peer review of "Humanized Ovarian Cancer Patient-Derived Xenografts for Improved Preclinical Evaluation of Immunotherapies"

_cancers, 2022, doi:10.3390/cancers14133092_

Round 1

Reviewer 1 Report

In this manuscript, the authors have established a humanized ovarian cancer xenograft mouse model to help improve immunotherapy efficacy in the future.

I think this established model is very interesting. I believe most readers will expect this model to contribute towards assessing the immunoresponse induced by cancer immunotherapy in the case of ineffectual therapy. Therefore, it may lead to more effectual cancer immunotherapy for ovarian cancer in the future.

Author Response

We would like to thank the reviewer for the positive evaluation of our manuscript.

Reviewer 2 Report

    This manuscript reports on the development of humanized CDX and PDX models of ovarian cancer. The authors establish that CD34+ cells that are isolated from human umbilical cord blood can effectively establish the human immune cell populations in NSG and NSGS mice. They performed extensive tissue profiling of immune subsets with 34 surface markers. These data are well-documented and clearly presented. The manuscript would be improved if the following points were addressed.

1.     The survival curve shown in Figure 5c is difficult to interpret. It appears that all three models are plotted as a single survival curve. The fact that there are five mice in the OV-90 group and only 2 mice in the PDX26 and PDX19 models makes this graph somewhat confusing.

2.     In Figure 5f, how were the error bars calculated when n=2?

3.     The tense and the verb/noun agreement are not correct in some places. The manuscript should be edited for English usage.

4.     The word “model” is repeated on line 518.

5.     Line 538, the text that reads “of more 12 weeks and longer” could be edited to “of more than 12 weeks”.

Author Response

Reviewer #2: This manuscript reports on the development of humanized CDX and PDX models of ovarian cancer. The authors establish that CD34+ cells that are isolated from human umbilical cord blood can effectively establish the human immune cell populations in NSG and NSGS mice. They performed extensive tissue profiling of immune subsets with 34 surface markers. These data are well-documented and clearly presented. The manuscript would be improved if the following points were addressed.

Response: We would like to thank the reviewer for the positive evaluation of our manuscript, and we appreciate the constructive comments.

Comment 1: The survival curve shown in Figure 5c is difficult to interpret. It appears that all three models are plotted as a single survival curve. The fact that there are five mice in the OV-90 group and only 2 mice in the PDX26 and PDX19 models makes this graph somewhat confusing.

Response: We agree with the reviewer that the survival curves may be difficult to interpret. As requested, we have updated Figure 5c and included three separate survival curves.

Comment 2: In Figure 5f, how were the error bars calculated when n=2?

Response: The reviewer raised a valid point and after the addition of three separate survival curves, we decided that Figure 5f is redundant and have removed the graphs.

Comment 3: The tense and the verb/noun agreement are not correct in some places. The manuscript should be edited for English usage.

Response: We understand that there might be some tense and verb/noun misplacements. However, the manuscript has been revised by expert academic proofreading services (scribendi) and the senior author, who is a native English speaker.

Comment 4: The word “model” is repeated on line 518.

Response: We have deleted the word “model” (now line 533) to avoid the repetition.

Comment 5: Line 538, the text that reads “of more 12 weeks and longer” could be edited to “of more than 12 weeks”.

Response: We have added the text according to the reviewer’s suggestion (page 15, line 554)

Reviewer 3 Report

The purpose of this research article was to set up a reproducible humanized mouse model for HGSOC in order to increase the translational impact of preclinical studies for immunotherapies in ovarian cancer. Additionally they studied the effect of the checkpoint inhibitor anti-PD1 nivolumab in these humanized mouse models.

Major comments:

1/ Materials and Methods: Please include details on patient derived tissues that could influence the TME in the patient and thus consequently also influence the outcome of the PDX models. E.g. Were the tumor tissues collected from treatment naïve patients, as chemotherapy has been shown to influence the TME? Were both samples from PDX26 and PDX 19 collected at similar disease stages in the patients? Did both tissues receive similar handling, more specifically cryopreservation before inoculation in mice or immediate injection after isolation from the patient? Did the tumor tissues originate from the primary tumor or metastatic sites, as differences between the TME of both sites have previously been observed? These factors could possibly explain the discrepancy between the results of PDX26 and PDX19 in these experiments.

2/ Figure 5: (A) what do the colored bars represent, is this the time of inoculation? Why was the PDX26 inoculated at an earlier timepoint compared to the other two models (week 2 versus week 9 after humanization)? Does the tumor growth and developing T-cells graphs in figure 5A represent the actual data or is this an example graph? Please rework this figure to produce a clear timeline of the experimental design.

3/ Figure 5 (B): Data from untreated control mice should be added to these graphs. In the first experiments, in the immune analysis of the humanized models, a similar upward trend was seen in hCD3 cells. Since there are no control mice included in these graphs, it is unclear if this increase of hCD3 can be related to the nivolumab treatment or not. Additionally, could you explain why the first immune analysis was performed on blood taken from week 8 while in two of the models the tumor inoculation only happened at week 9?

4/ Figure 5C: I think it is better to make separate survival graphs for all three models. Especially the PDX26 model in which the inoculation of the tumor cells occurred seven weeks before the other two models which subsequently means the PDX26 mice received treatment on week 10 after inoculation while the other two models received treatment 3 weeks after tumor inoculation. Therefore, the direct comparison of all models in one graph is not scientifically correct. The separation of the graphs would also increase readability.

5/ Results/Discussion: It would be recommended to include immune data of the humanized models after inoculation of the tumor cells. Ovarian cancer is known for its specific immunosuppressive microenvironment which should ideally be reproduced in the TME of the humanized models. Additionally comparison of the immune monitoring data should be correlated to the clinical setting in patients to prove translatability of this model.

6/ Discussion: I would suggest a more thorough discussion on the development of this specific humanized model, more specifically, comparing this model to already existing humanized mouse models (e.g. Hu-PBL-scid) using relevant references.

Minor comments:

1/ Materials and methods: could you please also mention at what time after humanization of the mice the digested patient derived tumors were inoculated? This is clearly stated in de CDX model but not the PDX model.

2/ Material and methods: Could you explain why bioluminescent imaging was done through intraperitoneal (IP) administration of the D-luciferin and measured after 10 minutes? In an article by Inoue et al. (https://doi.org/10.1007/s00259-008-1022-8) the comparison between subcutaneous and IP injection of D-luciferin was made where it was shown that IP injection of D-luciferin may overestimate the activity  of tumors residing in the peritoneal cavity. Moreover activity of the IP tumor starts immediately after IP injection of D-luciferin and not after 10 minutes. In contrast SC injection of D-luciferin offers more consistent results.

3/ Results (3.1): could you explain the choice to start the blood sampling analysis only at eight weeks after the CD34+ cells were injected? Why was this not also done earlier?

4/ Results: Could you mention the amount of mice that were used (n=…) in all figures?

5/ Results: in figure 1 (specifically C&D)  the production of human leukocytes in mouse injected with CD34+ cells between two mice strains (NSG vs NSGS) was compared. This comparison is clearly made in figure 1c for % of human CD45 cells, however this reviewer would suggest alterations for fig 1D so the comparison between the NSG and NSGS model is also addressed (now the NSG model data is put in supplementals). Suggestions would be:

-        the use of a similar graph as fig 1C in fig 1D.

-        or put both NSG and NSGS data on one bar graph with use of different colors.

6/ Results (3.2): could you explain why the analysis of the hematopoietic cells in the humanized mice was only performed 16 weeks after injection of the CD34 cells? Do you have any data on the different cell populations before this time point (this may also be relevant since you administer the anti-PD1 therapy from week 12 onwards)?

7/ Results (3.3 and afterwards). Could you specify  which type of model you have decided to use in all following studies? NSG vs NSGS (also mention this in the figure legends) as well as which concentration of busulfan and amount of CD34 cells that were selected for use in these experiments.

8/ Results: Figure 4A: in the figure, the number of mice used is n=7 while in the legend the number specified is n=6.

9/ Results: figure 5E: could you mention from which PXD model these fluorescent imaging originates (PDX26 or PDX19)?

10/ Results: Figure 6C: Please disclose at what time point the analysis of the tumor immune cells occurred. As the PXD26 model was inoculated at an earlier time compared to the other two models, could this be the explanation for the higher amount of infiltrated immune cells?

Round 2

Reviewer 3 Report

The authors have responded to all questions, the manuscript has been improved. I have no further questions.